# Novel *COX11* Mutations Associated with Mitochondrial Disorder: Functional Characterization in Patient Fibroblasts and *Saccharomyces cerevisiae*

**DOI:** 10.3390/ijms242316636

**Published:** 2023-11-23

**Authors:** Chenelle A. Caron-Godon, Stefania Della Vecchia, Alessandro Romano, Stefano Doccini, Flavio Dal Canto, Rosa Pasquariello, Anna Rubegni, Roberta Battini, Filippo Maria Santorelli, D. Moira Glerum, Claudia Nesti

**Affiliations:** 1Department of Biology, University of Waterloo, Waterloo, ON N2L 3G1, Canada; cacarongodon@uwaterloo.ca; 2Department of Neurosciences, Psychology, Drug Research and Child Health (NEUROFARBA), University of Florence, Viale Pieraccini, 6, 50139 Florence, Italy; stefania.dellavecchia@fsm.unipi.it; 3Molecular Medicine for Neurodegenerative and Neuromuscular Disease Unit, IRCCS Stella Maris Foundation, 56128 Pisa, Italy; stefano.doccini@fsm.unipi.it (S.D.); flavio.dalcanto@fsm.unipi.it (F.D.C.); anna.rubegni@fsm.unipi.it (A.R.); filippo3364@gmail.com (F.M.S.); cla_nesti@yahoo.it (C.N.); 4Experimental Neuropathology Unit, Division of Neuroscience, Institute of Experimental Neurology, IRCCS Ospedale San Raffaele Scientific Institute, 20132 Milan, Italy; romano.alessandro@hsr.it; 5Neuroradiology, IRCCS Stella Maris Foundation, 56128 Pisa, Italy; rosa.pasquariello@fsm.unipi.it; 6Developmental Neurology and Neurorehabilitation Unit, IRCCS Stella Maris Foundation, 56128 Pisa, Italy; rbattini@fsm.unipi.it; 7Department of Clinical and Experimental Medicine, University of Pisa, 56126 Pisa, Italy; 8Waterloo Institute for Nanotechnology, University of Waterloo, Waterloo, ON N2L 3G1, Canada

**Keywords:** COX11 mutation, mitochondrial diseases, yeast model, COX11 protein structure

## Abstract

Genetic defects in the nuclear encoded subunits and assembly factors of cytochrome c oxidase (mitochondrial complex IV) are very rare and are associated with a wide variety of phenotypes. Biallelic pathogenic variants in the COX11 protein were previously identified in two unrelated children with infantile-onset mitochondrial encephalopathies. Through comprehensive clinical, genetic and functional analyses, here we report on a new patient harboring novel heterozygous variants in *COX11*, presenting with Leigh-like features, and provide additional experimental evidence for a direct correlation between COX11 protein expression and sensitivity to oxidative stress. To sort out the contribution of the single mutations to the phenotype, we employed a multi-faceted approach using *Saccharomyces cerevisiae* as a genetically manipulable system, and in silico structure-based analysis of human COX11. Our results reveal differential effects of the two novel *COX11* mutations on yeast growth, respiration, and cellular redox status, as well as their potential impact on human protein stability and function. Strikingly, the functional deficits observed in patient fibroblasts are recapitulated in yeast models, validating the conservation of COX11’s role in mitochondrial integrity across evolutionarily distant organisms. This study not only expands the mutational landscape of COX11-associated mitochondrial disorders but also underscores the continued translational relevance of yeast models in dissecting complex molecular pathways.

## 1. Introduction

Cytochrome c oxidase (COX) is the terminal electron acceptor of the mitochondrial respiratory chain (MRC). Being a multisubunit enzyme, its biogenesis is a complex and finely regulated process, involving the assembly of protein subunits encoded by both nuclear DNA and mitochondrial DNA (mtDNA) [1,2,3]. COX11 is an integral protein of the inner mitochondrial membrane displaying high levels of evolutionary conservation [4,5], and has an essential role in the early steps of COX assembly, aiding in the insertion of copper into the enzyme [6,7].

Several COX mutations have been modelled in the yeast *Saccharomyces cerevisiae*, which has long been a very important tool for gaining relevant information on COX biogenesis and function. It has been shown that many COX11 residues conserved over evolution (including residues within the copper-binding domain) tolerate severe amino acid changes with no discernible effect on COX11 function, while others are essential for COX11 stability and function, or both [6]. Moreover, a subset of *cox11* petite mutants show a specific sensitivity to millimolar quantities of hydrogen peroxide (H_2_O_2_) [8]. While there have been several follow-up studies of the redox-related behaviour of COX11, the molecular basis for the peroxide sensitivity is still not understood [9,10]

In humans, several mitochondrial disorders caused by mutations in COX assembly factors have been described, presenting with a variety of phenotypes with varying degrees of severity ranging from skeletal myopathies, cardiomyopathies and encephalopathies to Leigh syndrome (LS), with the nuclear genes *SURF1*, *SCO1* and *SCO2* being more frequent [11,12,13,14,15]. Recently, *COX11* has been identified as a new disease-gene, with the description of biallelic mutations in two unrelated patients presenting with infantile-onset mitochondrial encephalopathy [16]. There has been relatively little study of COX11 in humans, although a recent study found that human COX11 is involved in control of heme A biosynthesis, as well as coordinating copper transfer to the assembling cytochrome oxidase complex [3]. Interestingly, a role for COX11 was recently identified in association with RANBP2 mutations in autosomal dominant acute necrotizing encephalopathy [17].

Here, we report a new case of *COX11*-associated mitochondrial disorder, identified using next-generation sequencing (NGS) methodologies in a child presenting with developmental delay, hypotonia, dyskinetic movements, and cerebellar and pyramidal signs. Combining morphological and biochemical investigations in muscle and cultured primary cells from the proposita with studies in the homologous yeast mutant strains, we demonstrate that there is a direct correlation between COX11 protein expression and the sensitivity to peroxide-induced oxidative stress, thus suggesting possible pathogenetic mechanisms at least for the *COX11* mutations identified in our study.

## 2. Results

### 2.1. Clinical Description

The case is an 11-year-old girl followed by our center since the age of 17 months. She was born to non-consanguineous parents by C-section because of breech presentation. Her family history was negative for neuropsychiatric disorders. Auxological parameters at birth were length 44 cm (1st percentile) and weight 2450 g (1st–2nd percentile). She could sit alone at 12 months and crawl and she took her first steps at 20 months. At 3 years, she produced lallations and she spoke single words at age 5. At the first neurological evaluation, at 17 months, she presented with hypotonia, global developmental delay, important cerebellar signs (balance difficulties, head and truncal titubation, intentional tremor especially of the upper limbs and dysmetria) and blurred pyramidal signs (brisk deep tendon reflexes and ankle clonus). The clinical picture progressed over time. She used a walker to move from the age of 3 years until the age of 6, when, due to the significant increase in muscle tone, walking became more difficult and baclofen therapy was required (up to a dosage of 75.5 mg at the age of 11 years). The MD CRS 4-18 Movement Disorder Scale administered at age 11 showed an index score of 0.68 corresponding to a severity class of 4 for the general assessment (which includes the assessment of motor function, verbal and oral function, self-care and attention/alertness) and an index score of 0.86 corresponding to a severity class of 5 for movement disorder assessment [18]. From the age of 5 years, the child presented lower limb and trunk dystonias and oro-buccal dyskinesias (spastic–dyskinetic tetra-paresis), as well as exotropia and pale optic papilla. From the age of 7 years and 6 months, episodes of apnea/hyperpnea and sleep disturbances characterized by crying, rigidity and pedal movements of the lower limbs lasting up to an hour appeared. Prolonged video-electroencephalogram (EEG) monitoring was performed, recording numerous awakenings during sleep characterized by psychomotor agitation with rhythmic movements of the lower limbs or by crying and startle-like movements, without a clear critical EEG correlate. These episodes responded positively to nitrazepam (1.25 mg as needed). At the last evaluation, the head circumference was 50 cm (<2° pc). Brain MRI showed progressive neurodegeneration, characterized by relapsing and remitting hyperintense lesions in T2, hypointense in T1 and with restricted diffusion in the acute phase, located in the peridentate white matter, mesial nuclei of the thalami, splenium of the corpus callosum, lenticular nuclei and mesencephalic tegmentum. These lesions were associated with reduced grey-matter/white-matter contrast in the cerebellar hemispheres and increased lactate peak in the cerebellum and lenticular nuclei. Figure 1 shows the neuroradiological picture over time.

At 3 years of age, EEG recording showed posterior slow activity triggered by eye closure and slow activity in sleep without paroxysmal abnormalities. The latest EEG, at 10 years and 8 months of age, showed posterior slow activity triggered by eye closure, and slow wave sequences and paroxysmal abnormalities in the middle vertex and centro-parietal regions; the paroxysmal abnormalities were activated by sleep, and during sleep the presence of diffuse rapid activity can be observed.

ECG, nerve conduction study, ophthalmic examination with fundus oculi exploration, visual evoked potentials and electroretinogram performed during the second and third years of life were normal. A repeated eye examination at the age of 6 years showed a pale optic papilla bilaterally and exotropia. Acoustic evoked potentials showed no truncencephalic responses for stimulation at 70 dBHL bilaterally.

A routine blood test showed a slight elevation of liver transaminases and iron-deficiency anemia. Analyses of acylcarnitines, sialotransferrins and lysosomal enzyme activity (arylsulfatase A, β-hexosaminidase, galactocerebrosidase and β-galactosidases) were normal. Plasma amino acids analysis showed an increase in alanine, valine and isoleucine, while urinary organic acids analysis showed an increase in lactic acid, pyruvic acid, ethylmalonic acid, succinic acid, fumaric acid, glutaric acid, 2-OH-glutaric acid and ketoglutaric acid (Appendix A).

Table 1 summarizes the genetic, clinical and neuroradiological features of our patient and the two other cases with mutation in *COX11* described in the literature [16].

### 2.2. Genetic Analysis

MitoExome sequencing was performed on the patient and her parents after obtaining informed consent. Bioinformatic analysis only revealed two heterozygous mutations in *COX11* (NM_004375.5), c.739C>A (p.P247T) inherited from the father and c.766dupA (p.T256Nfs*8) inherited from the mother. The missense mutation is absent in gnomAD polymorphic dataset (version v4.0.0, http://gnomad.broadinstitute.org/, accessed on 22 September 2023) and presents in silico evidence of pathogenicity (Appendix A), whereas the frameshift mutation has a frequency of 4 × 10^−6^ with no homozygotes reported in gnomAD. Following ACMG guidelines, they can be classified as a variant of uncertain significance (PM2, PM3) and likely pathogenic variant (PM2, PM3, PVS1), respectively. The frameshift mutation c.766dupA (p.T256Nfs*8) is located in the final exon of the gene and is predicted to produce a protein with a shorter C-terminal region, replacing the last 21 amino acids with seven alternative residues. Both affected residues showed a high degree of evolutionary conservation (Appendix A). COX11 cDNA analysis on dermal fibroblasts identified a single transcript presenting with the two heterozygous mutations, as verified by Sanger sequencing (Figure 2A), suggesting that both mutations are expressed in mRNAs that are not subject to nonsense-mediated decay.

### 2.3. Muscle and Cellular Studies

Histological analyses on patient skeletal muscle showed minimal nonspecific changes with the absence of ragged red fibers and normal staining for COX (Appendix A). MRC activities were in the normal range (Appendix A). Western blot (WB) analysis in muscle showed a severe reduction in COX11 protein (Figure 2B).

Cultured fibroblasts obtained from a skin punch biopsy were used to evaluate the bioenergetic features resulting from the COX11 defect. Micro-oxigraphy analyses revealed a normal respiration profile compared to controls, but with higher values in every step of oxygen consumption rate (OCR) recording (Appendix A). A specific mitochondrial complex IV enzyme activity did not show significant alterations, whereas the activity of citrate synthase (CS), a marker of mitochondrial proliferation, was significantly higher, consistent with an increased mitochondrial abundance. By further normalizing for CIV/CS activity, we detected a partial CIV dysfunction (about 68% residual activity) associated with compensatory mitochondrial proliferation, which was overlooked by considering only the enzymatic activities normalized to protein content (Figure 2C–E). These observations could explain the micro-oxygraphic trace, which relies on mitochondrial mass rather than on cell number.

Moreover, cultured dermal fibroblasts showed a higher susceptibility to oxidative stress, with a significant increase in reactive oxygen species (ROS) production both in basal conditions and after H_2_O_2_-induced oxidative damage (Figure 2F).

### 2.4. Mutation Analysis and Characterization in Yeast

Given that the analysis of patient muscle and fibroblasts suggested the presence of a subtle COX defect with an increased sensitivity to peroxide-induced oxidative stress, we decided to take advantage of the yeast model system to provide further insight into the molecular consequences of the novel COX11 mutations. Guided by a multiple sequence alignment of COX11 proteins from a broad array of eukaryotes, as well as several relevant prokaryotes, we used site-directed mutagenesis to generate *Saccharomyces cerevisiae* strains bearing the homologous mutations. While the p.P247T mutation readily translated to the p.P238T mutation in yeast, the p.T256Nfs*8 mutation was not as straight-forward to replicate (Appendix A). The patient’s frameshift mutation, which affects an amino acid that is not conserved, leads to non-conservative amino acid replacements in half of the amino acids comprising the final (10th) beta strand. Because of the considerable differences in nucleotide sequence between human and yeast, a direct translation of the patient mutation was not possible, especially because the resultant amino acid sequences between the two species would be divergent and the yeast COX11 contains a much longer C-terminal region (50 amino acids). We therefore chose to generate a premature stop codon at tyrosine 250 (Y250*), a residue shown to be important for COX11 function [6]. This variant predicts a truncated COX11 mRNA, likely mimicking the potential transcript created by the frameshift mutation detected in our patient.

#### 2.4.1. Respiratory Competence Is Maintained in the P238T Mutant

The P238T and Y250* mutant constructs were generated on both episomal and centromeric (CEN) plasmids and transformed into the well-studied W303ΔCOX11 strain. Following growth on glucose, replica plating to media with the non-fermentable carbon source ethanol and glycerol (EG) revealed robust growth for the P238T mutant, demonstrating respiratory competence in a haploid background. In contrast, the Y250*mutant was non-viable on EG (Figure 3A), displaying a complete respiration deficiency. Because the respiratory phenotype for the mutants was consistent, regardless of plasmid type, further characterization was carried out exclusively using the mutant constructs in the CEN plasmid, which at a copy number of 1–2 better reflects normal COX11 expression levels. As expected, the respiratory growth patterns were corroborated by the levels of specific COX activity in the mutants, with P238T having a similar COX activity as the strain expressing the wild-type COX11 and the Y250* mutant displaying only background levels of COX activity (Figure 3B).

#### 2.4.2. Stable COX11 Expression in Both Yeast Mutants

Based on the respiratory phenotypes of the two mutations in yeast, it was clear that the P238T allele conferred wild-type levels of respiration, while the Y250* allele failed to confer any respiratory capacity on W303ΔCOX11. To further characterize the two mutant strains, we used Western blotting of isolated mitochondria to determine the steady-state levels of several ‘marker’ COX subunits, COX2 (mtDNA-encoded) and COX4 (nDNA-encoded). As shown in Figure 3C, mitochondrially encoded COX subunits (COX1 and COX2) are undetectable in the Y250* mutant, along with reduced levels of COX4, similar to what is seen with the *cox11* knock-out. This result is consistent with the lack of detectable COX activity and the inability to grow on non-fermentable carbon sources. In contrast, but in keeping with the respiratory competent phenotype, the P238T strain displayed wild-type levels of all COX subunits and of the COX11 protein itself. Interestingly, we observed a very weak COX11 signal in the mitochondria isolated from the Y250* strain, migrating at approximately 25 kDa, which is the predicted size for the truncated protein. The presence of reduced, yet detectable, truncated COX11 presents an interesting anomaly, given that most protein truncations do not usually result in detectable steady-state levels of protein.

#### 2.4.3. Peroxide Sensitivity Phenotypes Correlate with Respiratory Competence

Peroxide sensitivity has been associated with the loss of either COX11 or SCO1 in yeast [8], although the molecular basis for this phenotype remains unresolved. The majority of *cox11* mutants that are respiration deficient also display peroxide sensitivity, though some variations on this have been reported. Using our standard peroxide sensitivity assay, the P238T mutant remains resistant to exposure to 6 mM hydrogen peroxide, whereas the Y250* strain displays a sensitivity similar to that seen for ΔCOX11 (Figure 3D).

### 2.5. Structure-Based Analysis of Human COX11 Mutations

The three-dimensional (3D) structure of human COX11 shows a β-strand fold organized into a homodimeric structure with an antiparallel arrangement forming a binuclear copper cluster at the dimer interface (Figure 4A). The N-terminal transmembrane helices (shown schematically in lower Figure 4A) anchor the dimer to the inner mitochondrial membrane and orient the two copper ions toward the membrane to facilitate the copper transfer process. Missense and frameshift mutations reported in this study (namely, p.P247T and p.T256Nfs*8) as well as the previously reported missense mutation p.A244P [16] are located at a distance from the copper binding site and were modeled in silico.

All three mutations are predicted to alter the normal structure of the COX11 protein (Figure 4B) and to affect the thermodynamic stability of both the COX11 monomer and homodimer (Figure 4C). Among the three mutations, T256Nfs*8 is predicted to cause the greatest changes in thermodynamic stability (ΔΔG monomer, T256Nfs*8: 12.99 ± 2.27 kcal/mol > P247T: 1.841 ± 0.002 kcal/mol > A244P, 0.93 ± 0.04 kcal/mol; ΔΔG homodimer, T256Nfs*8: 9.81 ± 1.35 kcal/mol > P247T: 3.40 ± 0.04 kcal/mol > A244P, 2.18 ± 0.15 kcal/mol).

Finally, the impact of mutations on the flexibility and dynamics of the COX11 homodimer was evaluated. Although minor changes (<4 Å) in the fluctuations of individual amino acids occur along the protein chains, significant alterations in protein flexibility and fluctuations were observed in the region of chain A where the mutated residues were located (Figure 4D), confirming that the mutations destabilize the COX11 homodimer.

## 3. Discussion

Over the past several years, NGS technologies have significantly improved the molecular diagnosis of rare genetic disorders such as mitochondrial diseases, highlighting their genetic and phenotypic heterogeneity. An exemplary case is COX deficiency, providing valuable information on the essential role of cofactors and on the presence of adaptative mechanisms.

The present work supports the disease-causative role of the mutations in *COX11* found in a girl with Leigh-like features. The disease onset seems to be slightly later in this case, being in early infancy, as compared to the previously reported cases with congenital onset [16]. The disease progression also seems to be milder, lacking respiratory distress, feeding problems and epilepsy. However, cognitive and motor impairment also appear to be significant in our case as in the previous children.

The neuroradiological phenotype of our patient showed a relapsing–remitting course of the lesions in the absence of widespread cerebral white matter involvement and cortico-subcortical atrophy. Brain MRS showed an increase in the lactate peak (lenticular nuclei and cerebellar white matter) as in one of the two previously described cases.

Although RC activities in skeletal muscle were within the normal range, studies in skin fibroblasts suggested a slight CIV reduction, detectable only by normalizing mitochondrial activity with the mitochondrial mass (Figure 2E). We identified a higher susceptibility to ROS, in concordance with previously published data [16], and future experiments will assess the effects of CoQ_10_ supplementation in our patient.

To gain a better understanding of the underlying pathogenicity, we took advantage of the yeast model. While the P238T mutation did not result in any detectable problems associated with COX activity or peroxide resistance, the Y250* mutant (likely mimicking the effect of the p.T256Nfs*8 mutation, the second allele found in our patient) showed an inability to assemble a functional COX, with the resultant degradation of several COX subunits (COX1, COX2 and COX4). The Y250* mutation also conferred a sensitivity to exogenous H_2_O_2_ exposure. Based on the complete loss of COX function in the Y250* mutant, our data suggest that the p.T256Nfs*8 mutation in *COX11* has a major contribution to the clinical symptoms associated with the mitochondrial disease in this patient. This is reminiscent of the compound heterozygote *SCO2* mutations described by Papadopoulou and colleagues [19], for which only one of the two mutations gave rise to a discernible respiratory deficiency in yeast [20]. The respiratory competent phenotype of the P238T COX11 mutant suggests a hypomorphic allele, based on the reduced thermodynamic stability of both the monomer and homodimer observed in the protein 3D models, and could explain the lack of a marked defect in MRC activity in patient fibroblasts and the relatively milder phenotype of our case, with a functional p.P247T COX11 compensating for the lack of COX11 function associated with the p.T256Nfs*8 allele. The detection of COX11 in the patient, migrating at the same molecular weight as observed in the control samples, further supports the notion that the p.P247T COX11 is stably expressed in the patient. The complete loss of function due to the truncation of the COX11 protein in the last β-strand suggests that the C-terminus is required for the structural integrity of COX11 in yeast, which is supported by the greatly reduced steady-state levels of the protein in the Y250* mutant. This hypothesis is corroborated by in silico modeling of the *COX11* variants, which confirms the major impact of the frameshift mutation on protein thermodynamic stability. Whether the C-terminus contains residues critical for COX11 function is currently under investigation and, given the difference in C-terminal length between the yeast and human COX11s, these experiments will shed further light on the structure–function relationships of this critical mitochondrial protein.

In summary, the present case supports the causative role of *COX11* biallelic mutations in mitochondrial encephalopathies, highlighting the association between COX deficiency and the remitting–relapsing course of the neuroimaging features, which deserve clinical attention and imply cautious neuroimaging follow-up. Moreover, the ability to separate mutant alleles in the yeast model system reiterates the value of utilizing yeast to study inherited human COX deficiencies.

## 4. Materials and Methods

### 4.1. Analyses on Patient’s Tissues

Routine morphology and histochemical stains for oxidative metabolism in muscle biopsy were performed according to standard protocols. MRC activities in skeletal muscle homogenate were measured using an existing spectrophotometric method [21]. Primary fibroblasts were obtained from diagnostic skin biopsy.

Mutations in the mtDNA and in *POLG* gene were excluded prior to this study. Genomic blood DNA from the patient was extracted by standard methods and analyzed using a multigene panel encompassing the coding regions of 1172 nuclear genes encoding the “MitoExome” (list of genes available upon request). The identified pathogenic variants in *COX11* (NM_004375.5) were confirmed by Sanger sequencing and were tested for segregation in the parents. Total RNA was isolated from skin fibroblasts, reversely transcribed to cDNA and amplified by PCR (primers available on request) following standard procedures.

WB analysis in muscle biopsy homogenate used a rabbit polyclonal anti-COX11 antibody (Biorbyt, Cambridge, UK; dilution 1:500) and GAPDH (Abcam, Cambridge, UK; dilution 1:5000) as the control.

OCR and ROS were measured in adherent fibroblasts using modified methodologies previously reported [22].

### 4.2. Generation of Yeast Mutants

Pairwise sequence alignments were performed using EMBL-EBI’s EMBOSS Needle (Clustal W) to determine the homologous sites for p.P247T and p.T256Nfs*8 in *S. cerevisiae* (Appendix A). Yeast proline 248 (CCA) was mutated to encode threonine (ACA) with the following primer set, 5′-GATTTTGCTAGCGAT**A**CAGCGATG-3′, and 5′-CTG**T**ATCGCTAGCAAAATCTGGATC-3′ (mutation in bold), to model the premature truncation of COX11, a duplication of adenosine 749 resulted in yeast tyrosine 250 (TAC) becoming a stop codon (TAA) (mutagenic primers 5′-CATTA**A**CACATTCTTCAGAGCTCATTATGG-3′ and 5′-GAGCTCTGAAGAATGTG**T**TAATGTAAAATA-3′). TaKaRa Ex Taq polymerase (Clontech labs, ThermoFisher) was used to perform site-directed mutagenesis on pCOX11/ST3.5 [6], which contains *COX11* on a 1.2 kb fragment in YCpLac111. Successful mutagenesis was confirmed via Sanger sequencing at The Center for Applied Genomics (Hospital for Sick Children, Toronto, ON). The plasmids expressing the mutants were individually transformed into haploid W303ΔCOX11 (*a ade2-1 his3-1,15 leu2-3,112 trp1-1 ura3-1 COX11::HIS3*) [6].

### 4.3. Characterization of Yeast Mutants

Following prototrophic selection, each strain was purified and tested with a ρ^0^ tester strain to ensure the presence of an intact mitochondrial genome. To test for respiratory competence, strains were grown overnight at 30 °C on YPD (1% yeast extract, 2% peptone and 2% dextrose) and respiratory growth was examined by replica-plating onto a non-fermentable carbon source, YEPG (1% yeast extract, 2% ethanol 2% peptone, 2% glycerol), and scoring for growth at 30 °C. Plates were imaged using the Syngene GeneFlash. In order to assess peroxide sensitivity, strains were grown overnight in YPD and then exposed to either 0 mM or 6 mM hydrogen peroxide [6]. Strains were grown in YPGal (1% yeast extract, 2% peptone and 2% galactose) in preparation for mitochondrial isolation, as described by [6]. The protein content of mitochondrial isolates was determined spectrophotometrically at 750 nm using the Folin reagent [23].

For Western blotting, 20 μg of mitochondrial protein was separated using 12% or 15% polyacrylamide gels, the latter including 15% glycerol. Proteins were then transferred to a nitrocellulose membrane (TransBlot Turbo, Bio-Rad Laboratories, Mississauga, ON, Canada) and stained with Ponceau S to verify protein transfer and subsequently blotted with the following antibodies: α-POR1, 1:10,000 (Molecular Probes, Eugene, OR, USA); α-COX11, 1:1000 [8]; α-COX1, 1:500 (Molecular Probes); α-COX2, 1:10,000 (Abcam, Cambridge, UK); α-COX4, 1:1000 (Abcam). The secondary antibodies were either α-Rabbit (Abcam) or α-Mouse (Thermo Fisher, Waltham, MA, USA) at concentrations ranging from 1:5000 to 1:20,000.

Proteins were visualized with Clarity Western ECL Substrate and imaged using a ChemiDoc MP (BioRad).

COX activity was measured using 1% cytochrome *c* (Sigma, St. Louis, MO, USA) in 20mM Tris-HCl pH 7.0, which was reduced with sodium dithionite. With a final 0.08% concentration of reduced cytochrome *c* in 20 mM potassium phosphate pH 7.0, 10 μL of deoxycholate (0.5%)-solubilized mitochondria (1:1) was assayed by measuring the decrease in absorbance at 550 nm. The specific COX activity was determined using the cytochrome *c* extinction coefficient (29.5 mM^−1^cm^−1^). Three biological replicates were tested in triplicate and the data were averaged across all replicates, where the positive control (pCOX11) was set to 100%.

### 4.4. Human COX11 Protein Modelling

The crystal structure of the human COX11-COX11 homodimer (model 5, cluster 2; [24] generated using Modeller [25] and the *Sinorhizobium meliloti* COX11 crystal structure as a template (PDB code: 1SP0; [26] was employed to assess the effect of missense and frameshift mutations on human protein. The effect of missense mutations on the thermodynamic folding stability of COX11 homodimer/monomer (i.e., the difference in Gibbs free energy between wild-type and mutant protein, ΔΔG) was calculated using the FoldX tool (version 5.0; [27]). Before using the BuildModel function of FoldX, the wild-type COX11 structure was subjected to the RepairPDB function in FoldX. Five independent runs were performed for each mutation and changes within FoldX’s predicted error range of 0.5 kcal/mol were considered insignificant.

Modeller version 10.4 [25] was used to model the structural changes in human COX11 caused by the frameshift mutation. Five different models of the truncated COX11 were generated and evaluated. The stability function of FoldX was used to calculate the Gibbs free energy required to fold the wild-type and mutated proteins from their unfolded state (ΔG). The changes in thermodynamic stability caused by the frameshift mutation were estimated by calculating the difference between the ΔG of the mutant and wild-type proteins (ΔΔG = ΔGmutant − ΔGwild-type).

The protein flexibility of the wild-type and mutant COX11 homodimer was evaluated using the CABS-flex 2.0 server (http://biocomp.chem.uw.edu.pl/CABSflex2, accessed on 4 September 2023 [28]). The CABS-flex simulation was carried out using default parameters and the results were presented as root mean square fluctuation (RMSF). The open-source UCSF ChimeraX (version 1.4; Ref. [29]) was used for structural rendering and GraphPad Prism (version 8.0) for data analysis.

## Figures and Tables

**Figure 1 ijms-24-16636-f001:**
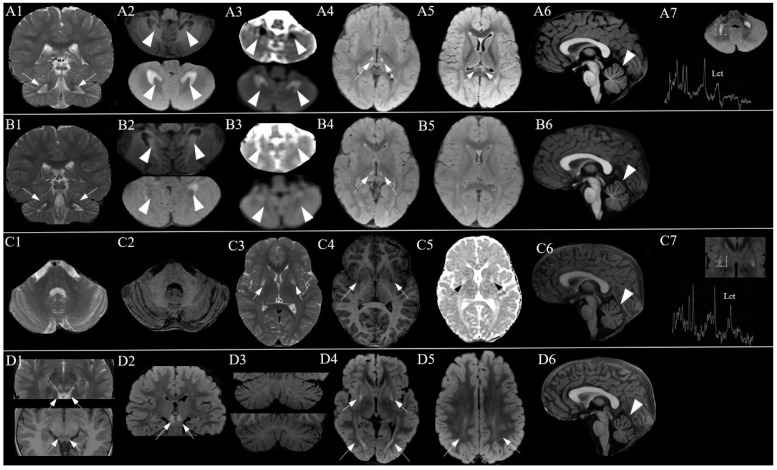
Brain MRI at 17 months (**A1**–**A7**). Coronal T2 FSE image (**A1**) and axial FLAIR image (**A2 below**) show bilateral and symmetrical hyperintensity of peridentate white matter which corresponds to hypointensity in T1 (**A2 above**) and restricted diffusion in ADC (apparent diffusion coefficient) (**A3**). Alterations in the mesial nuclei of the thalami (**A4**) and in the splenium of the corpus callosum (**A5**) show the same signal characteristics. Sagittal T1 image (**A6**) shows minimal enlargement of the perifolial space of the superior vermis. Magnetic resonance spectroscopy (MRS) showed increased lactate peak in the cerebellar white matter (**A7**). At 23 months (**B1**–**B6**), cerebellar white matter lesions (**B1**,**B2**) appear reduced and no longer characterized by restricted diffusion in ADC (**B3**) compared with the previous examination (**A1**–**A3**). Similarly, lesions in the mesial region of the thalami are seen (**B4**). The lesions in the splenium of the corpus callosum (**B5**) are no longer appreciable. The enlargement of the superior perivermian space is unchanged (**B6**). At 4 years (**C1**–**C7**), cerebellar white matter lesions are no longer appreciable (**C1**,**C2**). New lesions appear in the intermediate regions of the lenticular nuclei that appear hyperintense in axial T2 images (**C3**), hypointense in T1 (**C4**) and without restricted diffusion in ADC (**C5**). At this level, there is an observed increase in lactate peak at MRS (**C7**). Unchanged enlargement of the superior CSF perivermian space (**C6**). At 5 years and 10 months (**D1**–**D6**), new signal changes appeared in the mesencephalic tegmentum (**D1**,**D2**); coronal FLAIR image (**D3 above**) and coronal T1 image (**D3 below**) show reduction in grey-white matter contrast in the cerebellar hemispheres due to de/dysmyelination; axial FLAIR image shows T2 hyperintensity in the peritrigonal (**D4**) and supratrigonal white matter (**D5**) bilaterally. Sagittal T1 image (**D6**) shows unchanged enlargement of the superior CSF perivermian space.

**Figure 2 ijms-24-16636-f002:**
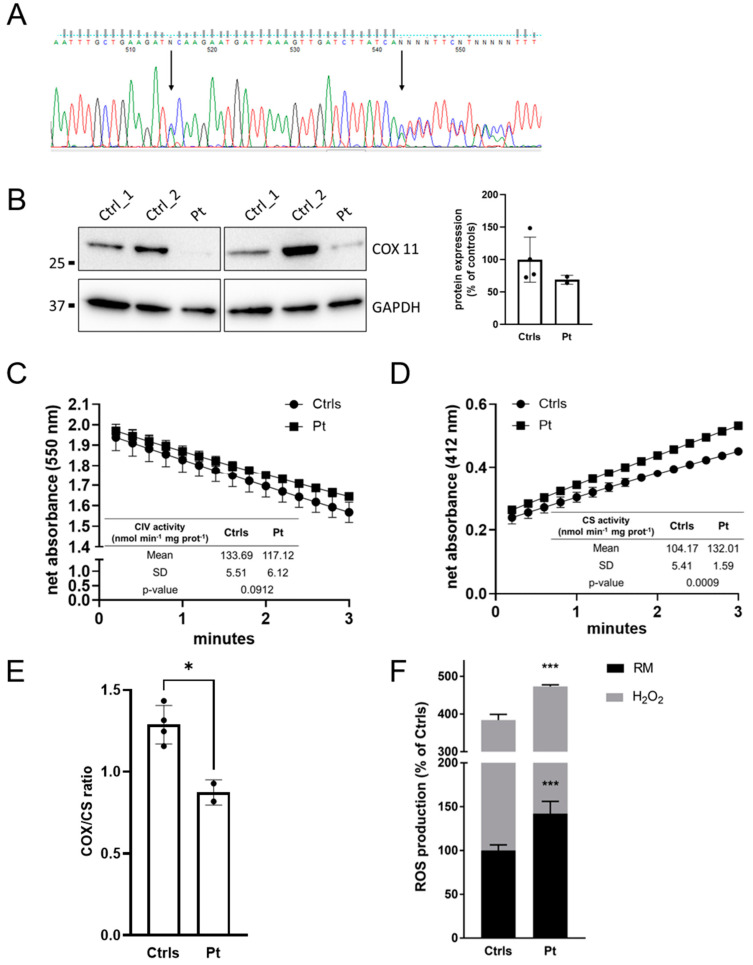
(**A**) Electropherogram showing *COX11* cDNA Sanger sequencing from patient fibroblasts, showing the presence of both heterozygous mutations c.739C>A and c.766dupA (indicated by arrows), meaning that the mutant transcript is not degraded by nonsense-mediated decay. (**B**) Western blot analysis of COX11 in muscle homogenates of patient and controls. Samples were resolved on a 8–16% tris-glycine gradient gel under reducing conditions, followed by transfer and then COX11-blotting. A low protein signal (normalized on GAPDH) was observed in patient as compared to both controls. A semiquantitative analysis of the immunoreactive bands was performed and data were graphed both as mean ± SD of two independent experiments and the scatter plot of each replicate. (**C**–**E**) Kinetic assessment of complex IV (CIV) and citrate synthase (CS) traces in patient and control fibroblast lysates. Cytochrome *c* oxidation at 550 nm (**C**) and TNB formation at 412 nm, coupled with the formation of CoA-SH by CS (**D**), were spectrophotometrically evaluated to calculate the ΔAbs/min. Although mitochondrial CIV enzyme activity did not show significant changes between the patient and controls, CS activities in the patient were significantly higher, more than 30%, consistent with an increased mitochondrial abundance (see tables inside panels). By further normalizing CIV activity to the activity of CS, we detected a partial CIV dysfunction associated with compensatory mitochondrial proliferation (**E**), which was overlooked by considering only the enzymatic activities normalized to protein content. Data were graphed as mean ± SD of two independent experiments. *t*-test with Welch’s correction was used to analyze differences between patient and controls group-values. (**F**) Evaluation of the susceptibility to oxidative stress in COX11 patient. Under both regular medium (RM) and stress conditions (H_2_O_2_), the patient’s cells produced a significantly greater amount of ROS compared with controls, indicating an increased susceptibility to oxidative stress. Data represent mean ± SEM of controls (n = 3) and the patient analyzed in technical triplicates. Statistical analysis was performed using an ordinary ANOVA test (one-way ANOVA). * *p* < 0.05; *** *p* < 0.001.

**Figure 3 ijms-24-16636-f003:**
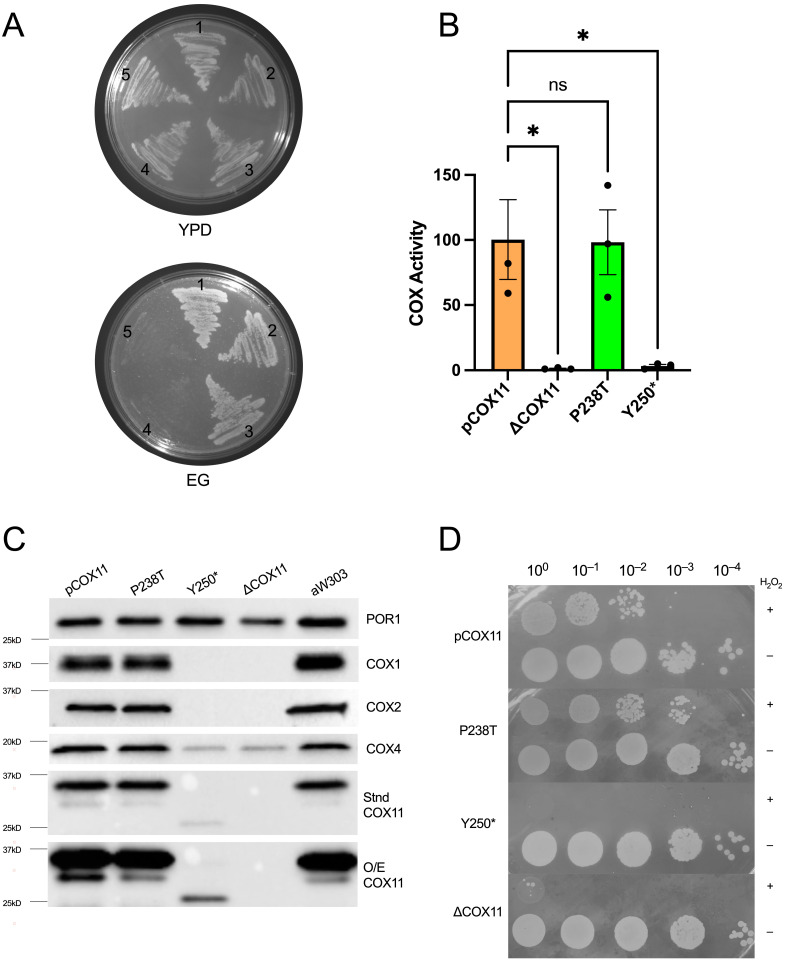
(**A**) Respiratory competence of yeast strains on non-fermentable media; (1) aW303, (2) pCOX11, (3) P238T, (4) Y250*, and (5) ∆COX11. (**B**) COX activity assay performed on mitochondria isolated from each strain. The average of three biological replicates is presented as a percentage of the positive control, pCOX11, activity, which was averaged and set to 100%. Error bars represent the standard error of the mean. An one-way ANOVA was performed using GraphPad Prism 10 where an asterisk indicates a *p*-value < 0.05. (**C**) Western blot analysis on 20 μg of protein from mitochondrial isolates. Blotted for COX subunits, COX11 and porin, which served as a loading control. Stnd COX11—blot at standard exposure. O/E COX11—same blot as standard COX11. However, the blot was overexposed to show faint bands. (**D**) Peroxide sensitivity assay. Each strain was grown for 2 h in the presence (+) or absence (−) of 6 mM H_2_O_2_ and plated on YPD media. Cells were grown at 30 °C for 36 h.

**Figure 4 ijms-24-16636-f004:**
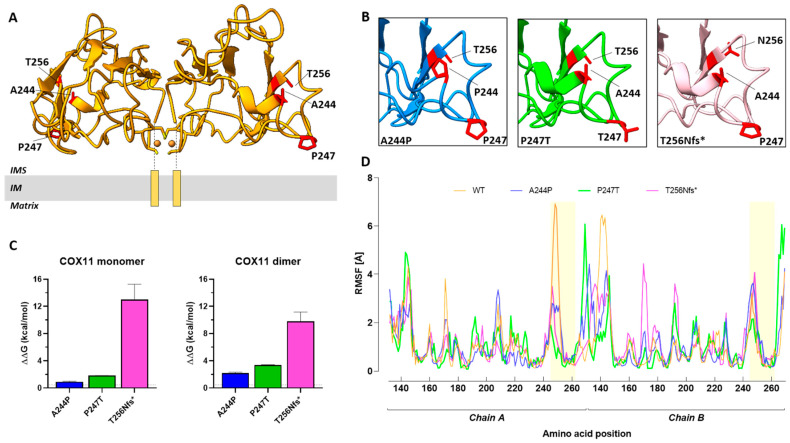
(**A**) Three-dimensional model of human COX11 homodimer. The human COX11 consists of an antiparallel homodimer anchored to the inner mitochondrial membrane (IM) by transmembrane helices (orange rectangles). This arrangement ensures that the two copper clusters of the COX11 dimer (orange spheres) face the membrane surface. The alanine residue at position 244, the proline residue at position 247 and the threonine residue at position 256 are highlighted in red. (**B**) Close views of the COX11 mutant structures around the different mutation sites. COX11 protein carrying the A244P mutation is shown in blue (left panel). COX11 protein carrying the P247T mutation is shown in green (middle panel). COX11 protein carrying the T256Nfs*8 mutation is shown in pink (right panel). The side chains of mutated residues are shown as sticks and highlighted in red. (**C**) Histogram of the calculated changes in free energy (ΔΔG) caused by A244P (blue), P247T (green) and T256Nfs*8 (pink) mutations in the COX11 monomer and homodimer. The dotted line indicates the FoldX prediction error (0.46 kcal/mol). Mutations are considered to be destabilizing when ΔΔG exceeds this value. (**D**) Root mean square fluctuation plots of wild-type (orange) and A244P (blue), P247T (green) and T256Nfs*8 (pink) mutant models of COX11 homodimer. The regions of the COX11 homodimer containing the mutated residues are highlighted in yellow.

**Table 1 ijms-24-16636-t001:** Genetic and phenotype-based findings in previously described patients and in the present case.

	Pt1	Pt2	Our Case
**Type of mutations**	Homozygous c.730G>C (p.A244P)	Homozygous c.35_36delinsG (p.V12Gfs*21)	Compound heterozygous c.739C>A (p.P247T), c.766dupA (p.T256Nfs*8)
**Last evaluation**	9 months	3 years	11 years
**Age of onset**	Congenital	Congenital	Early infancy
**First manifestation**	Sensorineural deafness (neonatal period)	Severe intrauterine growth retardation	Developmental delay, hypotonia, cerebellar and pyramidal signs
**Developmental delay**	Y	Y	Y
**Microcephaly**	NA	Y	Y
**Hypotonia**	Y	NA	Y
**Feeding and gastrointestinal disorders**	Poor feeding, nasogastric feeding	Gastrointestinal dysmotility and gastroesophageal reflux	N
**Respiratory** **distress**	Y (from 3 months)	N	N
**Visual disturbances**	Y	Y	Pale optic papilla and exotropia
**Dyskinetic movements**	Y (after 3 months)	N	Y (from 5 years)
**Cerebellar signs**	N	N	Y (from the first months of life)
**Pyramidal signs**	N	Y (increased tone and brisk deep tendon reflexes at 3 years)	Y (from the first months of life with progressive spasticity)
**Epileptic seizures**	Y (clonic seizures from the first months of life)	Y (focal seizures for 3 years)	N
**Sleep disorders**	NA	NA	Y (from 7.6 years)
**EEG**	NA	Generalized epileptiform discharges at 7 months	Posterior slow activity triggered by eye closure, slow wave sequences and paroxysmal abnormalities (activated by sleep) in the middle vertex and centro-parietal areas. During sleep, diffuse, unusual, rapid activity.
**CSF lactate**	↑	↑	NA
**Plasma lactate**	↑ intermittently	NA	NA (↑ at MRS and UOA analysis)
**PAA and UOA**	NA	Tandem mass spectrometry and urine gas chromatography were normal	Altered *
**Brain MRI**	At the beginning, it was normal. At 3 months, new lesions in the cerebellar peduncles. At 7 months, brain atrophy, dentate nucleus lesions, and diffuse WM disease.	At 1 year, bilateral basal ganglia hyperintensity, mild brain atrophy. MRS showed lactate peak in basal ganglia.	Relapsing and remitting lesions hyperintense in T2, hypointense in T1 and with restricted diffusion in the acute phase in the peridentate WM, mesial nuclei of the thalami, CC splenium, lenticular nuclei and mesencephalic tegmentum. ↓ of grey/white matter contrast in the cerebellar hemispheres. ↑ lactate peak in the cerebellum and lenticular nuclei.
**Death**	9 months	Still alive	Still alive

* See Appendix A. Y: yes. N: not present. NA: not available. CSF: cerebrospinal fluid. MRS: magnetic resonance spectroscopy. PAA: plasma amino acids. UOA: urine organic acids. MRI: magnetic resonance imaging. WM: white matter. CC: corpus callosum.

## Data Availability

All raw data presented in this study not reported in the text or in the Appendix A are available upon request.

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
