# Peer review of "Novel *COX11* Mutations Associated with Mitochondrial Disorder: Functional Characterization in Patient Fibroblasts and *Saccharomyces cerevisiae"

_ijms, 2023, doi:10.3390/ijms242316636_

Round 1
Reviewer 1 Report
Comments and Suggestions for Authors
This well-written study reports one patient with Leigh-like disease found to be compound heterozygous for novel COX11 variants. To my knowledge there is only one previous study of human mitochondrial disease associated with this gene (of 2 unrelated individuals), so this provides important evidence corroborating COX11 as a rare cause of autosomal recessive mitochondrial disease.
A clear and detailed clinical description is provided, including brain MRI findings (although note I am not a neuroradiologist and cannot critically evaluate Figure 1). The authors have helpfully compared findings in their patient with those in the two patients published by Ruis et al. (Table 1).
In addition, the authors have undertaken analysis of patient muscle and fibroblasts, extensive studies using a yeast model, and structural/molecular modelling, all of which add further scientific value.
Specific major comments:
· The authors should evaluate variant pathogenicity of the two COX11 variants using variant interpretation guidelines (Richards et al. 2015 PMID: 25741868 and ClinGen https://clinicalgenome.org/working-groups/sequence-variant-interpretation/). This may result in one or both variants being classified as variants of uncertain significance (VUSs); therefore, depending on the outcome of variant classification, the authors may need to modify their wording to be more cautious regarding the clinical significance of these variants; for example, if both variants cannot be classified as (likely) pathogenic, then the authors should state that their findings support a diagnosis of COX11-related disease, but that this cannot currently be confirmed/proven (or similar appropriate wording).
· Page 5, line 160: Muscle respiratory chain activities are said to be normal; please can this information be provided in supplementary material? Also, given the discussion around CIV activity in fibroblasts, was muscle CIV activity also normal when normalised against CS activity?
· Page 11, lines 321-323: Following on from my previous comment, these lines state that there is evidence of a slight CIV reduction in muscle, but this has not been presented in the results and so requires rectifying.
· Page 11, lines 330-346: This section of the discussion is currently misleading and requires rewording for a number of reasons. a) the p.T256Nfs*8 variant cannot alone be the underlying cause for the clinical systems since it was inherited from the proband’s asymptomatic mother; therefore, the text should be reworded to suggest that this variant has a more major contribution to the phenotype than p.P247T. b) When proposing that p.P247T may be a hypomorphic allele, it would be helpful to highlight evidence from this study supporting some functional defect of this variant (i.e. molecular modelling showing evidence of reduced stability of the COX11 homodimer). c) The final two sentences of this paragraph (lines 341-346) are somewhat contradictory as the authors suggest both an essential function associated with the C-terminus (which I would take to mean a functional domain or similar) and loss of thermodynamic stability (which could lead to absent or very low levels of stable protein – supported by the Y250* yeast studies); therefore, rewording is required to clarify what the authors believe is the most likely consequence of p.T256Nfs*8 (e.g. absent or severely reduced levels of protein, or perhaps a stable non-functional protein due to loss of critical C-terminal amino acids?). For clarity, it would also be helpful to state somewhere in the manuscript that c.766dupA p.T256Nfs*8 is located in the final exon of the gene, is not predicted to undergo nonsense-mediated mRNA decay (supported by the authors cDNA analysis), and is predicted to result in replacement of the C-terminal 21 amino acids by 7 alternative amino acids.
· As the previous publication of COX11-related mitochondrial disease (Ruis et al.) studied and discussed rescue of dysfunction in fibroblasts by CoQ10 supplementation, it is surprising that this has not been investigated here. The authors should at least comment that further work could be undertaken to assess this for their patient.
Minor comments:
· Page 2, line 1: “parentsby” is missing a space.
· Page 3, lines 121-126: Please state patient age when these EEGs were undertaken.
· Page 4, line 142: I can’t find any supplementary detail describing PAA and UOA results, although this is summarised in the text (lines 135-138).
· Page 5, line 169: Please state the percentage reduction in CIV/CS activity compared to controls.
· Page 6, lines 200-216: Why is this text italicised?
· Page 8, lines 245-247: Given that the Y250* mutant is not predicted to undergo nonsense-mediated mRNA decay (assuming this truncation is in the final exon of the yeast gene as it is in human) and is only 18 amino acids truncated compared to human COX11, the ability to generate some stable protein does not seem particularly surprising.
· Page 8, line 248: The heading “Premature truncation of COX11 leads to peroxide sensitivity” is misleading as this can likely be accounted for by the greatly reduced level of stable protein rather than any functional defect associated with the 25 kDa Y250* protein. Therefore, this should be reworded.
· Page 8, lines 250-252: This statement referring to peroxide sensitivity of other cox11 mutants should be referenced or removed.
· Page 9, Figure 3C: The bottom panel should be labelled “O/E COX11” not “Stnd COX11”.
· Pages 12-13, lines 354-433: some text is in italics and other text not, with no obvious reason for this difference.
· Supplementary figure S3: It would be helpful to include additional mammalian and vertebrate species in this alignment to help illustrate evolutionarily conserved amino acids in this region of the COX11 protein.
Reviewer 2 Report
Comments and Suggestions for Authors
Review: IMJS 2707863
Novel COX11 mutations associated with mitochondrial disorder: functional characterization in patient fibroblasts and Sac-3 charomyces cerevisiae
The authors describe in detail functional and genetic findings relating to mutations in mitochondrial gene COX11, in a patient with Leigh syndrome-like symptoms. The manuscript is well written and the experimental work sound. There are some minor changes required.
Results line 69
Spacing “parentsby”
Results line 89, Table 1
All abbreviations should to be spelled out in full the first time they are mentioned (EEG, CSF, etc).
Results line 200
Font and indent do not much the rest of the document.
Results line 288, Figure 4
The quality of this figure is lower than in the other figures and appears grainy when magnified, if it is possible, the authors may want to improve the image quality.
Materials and Methods lines 353, 369, 383, 411
Font and indent do not much the rest of the document.
Materials and Methods line 454
The data availability statement reads as follows:
“Data Availability Statement: In this section, please provide details regarding where data support-454 ing reported results can be found, including links to publicly archived datasets analyzed or gener-455 ated during the study. Please refer to suggested Data Availability Statements in section “MDPI Re-456 search Data Policies” at https://www.mdpi.com/ethics. If the study did not report any data, you 457 might add “Not applicable” here.”
This should be corrected.
